# A Multi-Elements Isotope Approach to Assess the Geographic Provenance of Manila Clams (*Ruditapes philippinarum*) via Recombining Appropriate Elements

**DOI:** 10.3390/foods10030646

**Published:** 2021-03-18

**Authors:** Eun-Ji Won, Seung Hee Kim, Young-Shin Go, K. Suresh Kumar, Min-Seob Kim, Suk-Hee Yoon, Germain Bayon, Jung-Hyun Kim, Kyung-Hoon Shin

**Affiliations:** 1Department of Marine Sciences and Convergent Technology, Hanyang University, Ansan 15588, Korea; ejwon@hanyang.ac.kr (E.-J.W.); bearpig4528@gmail.com (S.H.K.); hc12sook@gmail.com (Y.-S.G.); ksuresh2779@gmail.com (K.S.K.); 2Department of Botany, University of Allahabad, Allahabad 211002, India; 3Department of Fundamental Environmental Research, Environmental Measurement & Analysis Center, National Institute of Environmental Research, Incheon 22689, Korea; candyfrog@korea.kr (M.-S.K.); yoonsh1120@korea.kr (S.-H.Y.); 4IFREMER, Marine Geosciences Unit, F-29280 Plouzané, France; Germain.Bayon@ifremer.fr; 5Korea Polar Research Institute, 26 Songdomirae-ro, Yeonsu-gu, Incheon 21990, Korea; jhkim123@kopri.re.kr

**Keywords:** authentication, Manila clam, stable isotope, traceability, linear discriminant analysis

## Abstract

The increasing global consumption of seafood has led to increased trade among nations, accompanied by mislabeling and fraudulent practices that have rendered authentication crucial. The multi-isotope ratio analysis is considered as applicable tool for evaluating geographical authentications but requires information and experience to select target elements such as isotopes, through a distinction method based on differences in habitat and physiology due to origin. The present study examined recombination conditions of multi-elements that facilitated geographically distinct classifications of the clams to sort out appropriate elements. Briefly, linear discriminant analysis (LDA) analysis was performed according to several combinations of five stable isotopes (carbon (δ^13^C), nitrogen (δ^15^N), oxygen (δ^18^O), hydrogen (δD), and sulfur (δ^34^S)) and two radiogenic elements (strontium (^87^Sr/^86^Sr) and neodymium (^143^Nd/^144^Nd)), and the geographical classification results of the Manila clam *Ruditapes philippinarum* from Democratic People’s Republic of Korea (DPR Korea), Korea and China were compared. In conclusion, linear discriminant analysis (LDA) with at least four elements (C, N, O, and S) including S revealed a remarkable cluster distribution of the clams. These findings expanded the application of systematic multi-elements analyses, including stable and radiogenic isotopes, to trace the origins of *R. philippinarum* collected from the Korea, China, and DPR Korea.

## 1. Introduction

The Manila clam (*Ruditapes philippinarum*) is one of the most important shellfish species in fisheries because of its nutritional value, flavor, and low cost [1,2,3]. The cultivation of Manila clams has rapidly advanced in many countries due to rapid growth rates, fecundity, and ease of cultivation. The countries with high consumptions of clam, such as Italy and Korea, rely on domestic production as well as imports from other countries, such as Vietnam and China [4]. China is the main global exporter of clams, which mainly supplies the Asian market. Cultivation of the Manila clam *R. philippinarum* has thrived as seeds are traded and farmed among nations to counter increasing market demands that the indigenous population cannot meet. Over 4000 tons per annum of Manila clam seeds imported from China have been transplanted to coastal regions of Korea and are already genetically mixed, rendering their geographical origins difficult to determine based solely on genetic information [5,6].

As the global trade in seafood products increases, the importance of the authenticity to meet the demands of health-conscious consumers has grown [7]. The pressure from many countries for effective authentication and traceability tools to guarantee quality and safety is also increasing [6,8]. Factors involved in the shellfish industry, such as increased popularity of clam consumption, seafood globalization, and recent reports of contaminated food products, have led to consumer concerns regarding the quality, origin, traceability, and authenticity of many types of seafood, including clams [9]. Particularly, the origin (food authentication) is one piece of important information that enables consumers to predict or trust the quality of seafood [10]. Therefore, elemental profiling, stable isotope analysis (SIA), lipid profiles, and near-infrared spectroscopy are being considered for authenticating food products because they can overcome the limitations of genetic information (DNA barcoding) [5,11,12,13]. Among these approaches, SIA facilitates rapid and efficient tracing of unsafe products. Stable isotopes are natural ecological recorders that can provide valuable biogeochemical information about water, food, diet, travel, and even manufacturing processes [14]. Thus, differences in isotope ratios among aquatic organisms can reflect distinct geographical provenances, feed types, or habitats based on ecological niches [9,15]. In fact, SIA has assumed a leading role in determining the authenticity of food origins for producers and control agencies and has proven particularly valuable in tracing the origins of adulterated food [16,17].

Recent reviews of food traceability have summarized factors affecting stable isotope ratios of seafood items that can provide information for authentication [9,18]. The fractionation of ^2^H/^1^H and ^18^O/^16^O caused by evaporation, condensation, and precipitation, can help determine geographical origins, whereas that of ^13^C/^12^C due to different metabolic pathways in C_3_ and C_4_ plants can help determine dietary habits that can consequently indicate geographic origins and farming methods. The fractionation of ^15^N/^14^N caused by changes in agricultural practices through the food chain can also provide information on diets, which in turn can help indicate geographical origins by reflecting ecological niches, and bacterial fractionation of ^34^S/^32^S can be also used for marine geographical determinations associated with biogeochemical sulfur cycles. Sulfur isotope ratios (^34^S/^32^S) with little or no fractionation can also serve as indicators of origin, but these have not been investigated in detail from the perspectives of animal science and food authentication [19].

Continuous improvements in analytical technologies have led to the development of additional isotopic systems for provenance and environmental studies [20]. Unlike stable isotopes, radiogenic isotopes are not fractionated during biogeochemical processes and can thus serve as unique tracers for provenance studies. Strontium (^87^Sr/^86^Sr) is a radiogenic isotope that can deliver unique information to help trace the origins of water and food products [21]. A residence of ~3 million years, which is far longer than the mean mixing time of ~1500 years for all oceans, means that the ^87^Sr/^86^Sr content in seawater is relatively uniform (~0.7092) [22]. The isotope ratio of the rare radiogenic earth element neodymium (Nd) (^143^Nd/^144^Nd) can be useful for provenance studies [23]. Unlike Sr, Nd isotopes vary according to water mass, thus rendering seawater archives useful for tracing the origin of the water masses from which these isotopes were precipitated [24].

Recent studies have proven that SIA is a powerful verification and traceability tool that can trace food and thus alleviate customer concerns about whether a purchase in fact does originate from a declared source, help combat illegal trade, and authenticate various exports [9,16,25]. Furthermore, studies on the authenticity and geographical origins of edible products focused on processed food such as fruit juices, wines, honey, milk products, tea, coffee, and agricultural products including crops, meats, feeds and habitats (cows and swine) using SIA showed the commercial potential of this approach [17,26]. Commercial instruments such as SPEX^®^ (SamplePrep, LLC, Metuchen, NJ, USA) can evaluate ratios of stable C, O, N, H, and S isotopes to determine food product origins, and the suitability of SIA for tracing aquacultured products from various geographic locations has been discussed [17], but few studies have emphasized the applicability of these ratios to inferring seafood provenance [15]. In fact, unlike raw materials and some items that undergo specific processing in factories, such as juices, milk, or wines, informational elements and combinations are difficult to select because aquatic products were live before distribution or cooking. Thus, further study of the potential of isotopes to comprehensively trace food is warranted. The present study aimed to determine whether stable (C, N, O, H, S) and radiogenic (Sr, Nd) isotopes could help authenticate the geographical origins of clam samples and what elements can be suitable elements for authentication of three countries. For this, the most basic elements C and N were set as defaults, then we added five elements under various conditions and applied them to the linear discriminant analysis (LDA) analysis.

## 2. Materials and Methods

### 2.1. Sample Collection

In 2015, samples of Manila clams (Figure 1) were collected from the coastal areas of Democratic People’s Republic of Korea (DPR Korea) (*n* = 10), China (Dalian (*n* = 10) and Korea (*n* = 10). Samples of Korean clams were collected by researchers from three regions that were close to each other, but with different environmental characteristics: Yeosu Hwayang (*n* = 3), Yeosu Dolsan (*n* = 4), and Jinhae (*n* = 3). These were immediately sent to the laboratory, where they were weighed, measured, and grouped according to size.

The test tissue was the clam adductor muscle, which is easily distinguishable and has a long turnover rate. The samples were dissected and stored in at −80 °C before being crushed, homogenized using a mortar and pestle, and ground in a planetary ball mill (Fritsch GmbH., Idar-Oberstein, Germany).

### 2.2. Stable Isotope Analysis

Stable nitrogen isotopes were directly studied in sample homogenates without further processing. Lipids were extracted from sample homogenates using a 2:1 ratio of chloroform and methanol for carbon isotope analysis as described [15].

Stable isotopes were studied using a CHNS_O elemental analyzer (EuroVector S.p.A., Milan, Italy) connected to an Isoprime 100 isotope ratio mass spectrometer (Isoprime Ltd., Cheadle Hulme, UK). The carbon and nitrogen references were Vienna PeeDee Belemnite and atmospheric nitrogen (N_2_), respectively.

Ratios of stable hydrogen, oxygen, and sulfur isotopes in samples were measured using an Isoprime vision continuous flow isotope ratio mass spectrometer (GV Instruments Ltd., Manchester, UK) coupled to a Vario Microcube elemental analyzer (Elementar Analysensysteme GmbH., Langenselbold, Germany) for sulfur isotopes and a high-temperature pyrolysis conversion elemental analyzer for oxygen and hydrogen isotopes.

Isotopic ratios are presented as δ values (‰) relative to the standard Vienna Canyon Diablo Troilite (V-CDT) for sulfur and Vienna Standard Mean Ocean Water (V-SMOW) for oxygen and hydrogen, respectively.

Isotope ratios are reported per mille (‰) using standard delta notation. The stable isotope ratios of each element are expressed as values relative to international reference materials as follows (1):δ^13^C or δ^15^N (‰) = [(^13^C/^12^C or ^15^N/^14^N) _sample_/(^13^C/^12^C or ^15^N/^14^N) _standard_−1] × 1000;δX = [(R_sample_−R_std_)/R_std_] × 1000 (‰)(1)
where X = ^2^H or ^18^O or ^34^S, R = ^2^H/^1^H or ^18^O/^16^O or ^34^S/^32^S, and std (standards) = V-CDT for sulfur, and V-SMOW for hydrogen and oxygen.

Reference materials were IAEA-CH-3 (Cellulose, −24.724‰) for ^13^C/^12^C, IAEA-N-1 (Ammonium sulfate, 0.4‰) for ^15^N/^14^N, IAEA-S-1 (silver sulfide, −0.3‰), NBS-127 (barium sulfate) for ^34^S/^32^S (20.3‰), and IAEA-601 (benzoic acid, 23.14‰) for ^18^O/^16^O and IAEA-CH-7 (Polyethylene foil, −100.3‰) for ^2^H/^1^H. This working standard was analyzed at regular intervals (*n* = 10) in each run to control the repeatability in analysis. The analytical deviations of the standards were both < ± 0.2‰ for C and N. The analytical precision was within 0.5‰, 0.2‰, and 1.2‰ for sulfur, oxygen, and hydrogen, respectively.

### 2.3. Radiogenic Sr and N Isotope Analysis

Powdered samples (~50 mg, dry wt.) were digested with ultrapure concentrated HNO_3_ in vials on a hotplate at 120 °C. Strontium and neodymium were purified using conventional ion chromatography. Isotopes were measured at the Pôle Spectrométrie Océan (Plouzané, France) using a Thermo Scientific Neptune multi-collector ICP-MS. Mass bias in measured ^87^Sr/^86^Sr and ^143^Nd/^144^Nd values was corrected by an exponential law using ^86^Sr/^88^Sr = 0.1194 and ^146^Nd/^144^Nd = 0.7219. The ratios of the Sr and Nd isotopes were normalized using NIST SRM-987 and JNdi-1 standards, respectively. The Nd isotopes are described using epsilon notation (εNd), which corresponds to the deviation of the measured ^143^Nd/^144^Nd to the presently inferred composition of the bulk primitive Earth (or CHUR), using^143^Nd/^144^Nd_CHUR_ = 0.512630 [27].

### 2.4. Statistical Analysis

Data are presented as mean ± standard deviation (S.D.). Before statistical analysis, the normal distribution of data and homogeneity in variances were checked by Levene’s test. Significant differences of each isotope data according to countries and regions were analyzed by one-way ANOVA and its post hoc analysis following Tukey’s test. The Dunnett test was used for samples that have no normal distribution to compare groups. Linear discriminant analysis (LDA), which is one of the most popular supervised approaches for food authentication, characterization and adulteration detection [28], was used to find a linear combination of variables and calculate the discriminant function that best separates classes. To evaluate the LDA performance, predicted class separated by means of country was validated by internal leave-one-out cross-validation (LOOCV) method under the same data set [29,30]. After designating each country with a different color, samples for which the origins were not properly classified were indicated in bold italics with the color of the identified country. All the statistical analysis including LDA and cross-validation was performed with R software with Vegan package in R (v.3.6.1., R core team, 2019; v2.5-6) [31]

For LDA processing, data (stable isotope ratio) were transformed by Z-score normalization method with the following Equation (2):(2)$$A_{x^{\prime }}=\frac{A_{x}\;-\;\mu }{\sigma }$$
where *A_x_* and *Ak_x’_* stand for isotope ratio of the specific element, *μ* and *σ* represent the average and standard deviation values of each element’s isotope ratio, respectively. This formula allows the difference of isotope values between data, expressed to deviation from the average and set all isotope values to similar ranges, thus reducing the bias derived from the difference between each isotope values range.

## 3. Results and Discussion

Table 1 shows the average values of stable isotopes in Manila clams collected from sites in Korea, DPR Korea, and China. These are transformed and normalized by Z-score for statistical processing (LDA) (Table 2). The δ^13^C value for the clam sample from DPR Korea was −17.7 ± 0.4‰, and slightly lighter than those from Korea, indicating differences in particulate organic matter (POM), the composition of other potential foods and behaviors of the clams (Table 1) [32].

The wide range of δ^13^C values from −18.2 to −16.2‰ measured in clams from the same three countries can be explained if the POM consumed by clams differed in terms of carbon isotope values in various regions. The δ^13^C values of the Manila clams from China were relatively lighter (−17.7 ± 0.4‰) than those from Korea, but similar to that for bay scallops (−18.0 ± 0.8‰) from Bohai Bay, where the Manila clams were sourced (Table 1) [33]. It was even found that the wide range of δ^13^C values for clam muscle (−20.2–−16.4‰) collected from 50 sites in France found site-dependent differences and that additional information is required to produce more reliable distinctions [34]. In addition, differences in samples were smaller when sourced from the same, rather than different, countries, even when sourced from neighboring countries. The δ^13^C composition was similar in Manila clams from the Hwayang and Dolsan sites in Yeosu (−17.1 ± 0.02‰ and −17.1 ± 0.2‰), while those from Jinhae were distinct with very little variability (−16.3 ± 0.05‰). The δ^13^C values of clams from the two Yeosu sites were also similar, as the sites were in close proximity. The small δ^13^C differences in clams collected from the sites in Korea in the present study might reflect seasonal and/or dietary preferences [15], and that the distinct δ^13^C value for the Manila clam may be based on provenance. In fact, the δ^13^C values at the three sites in Korea were close to those of invertebrate suspension feeders in the intertidal zone adjacent to the sites where the clams were collected for this study [35].

The δ^15^N values of the Manila clams collected from the three countries were similar, but with significantly different ranges (*p* < 0.05, one-way ANOVA). In samples from Korea, the δ^15^N values of the Manila clam found in Yeosu Hwayang, Yeosu Dolsan, and Jinhae differed, from 8.7~10.1‰. In general, variations in stable nitrogen isotope composition help determine dietary shifts, differences in the trophic level of prey species, and variations in the baseline levels in food webs in ecological approaches. In particular, considerable fractionations have been compared between nitrogen (mean trophic increase of 2–4‰) and carbon (~1‰) during dietary assimilation [36] and used to determine trophic positions between preys and predators. This may seem completely irrelevant to the geographical authentication but allows the determination of trophic relationships and positions in complex food webs. Our recent study also showed that trophic information associated with organic sources and prey can discriminate habitats, as these values generally vary with environments and structured food relationships in ecosystems [36]. In this study, two different freshwater fish were good examples, showing that the stable nitrogen isotope ratio changes occurring seasonally along the nitrogen cycle can be an indicator of geographical origin [36]. It means that environmental characteristics that might affect nitrogen isotope ratios can be applied in the interpretation of geographic information. Watanabe et al. proposed that δ^15^N values in soft tissues of *R. philippinarum* could indicate anthropogenic eutrophication levels in tidal flat environments [37].

Like the δ^13^C values, the δ^15^N values of the Chinese Manila clam (8.8 ± 0.2‰) determined herein were similar to those described by Wu et al. [38] and Zhao et al. [3], namely, 5.1‰ and 8.0–8.6‰, for POM and the Manila clam, respectively. Due to an average enrichment of 3‰ over dietary δ^15^N, nitrogen isotopes are also particularly useful for placing animals in a trophic hierarchy [14,36]. The δ^15^N value of the DPR Korean clams was 9.0 ± 0.2‰, which, despite the geographic vicinity, was lighter than that of Korea. These trends (or differences) in carbon isotope values from Korea and DPR Korea might be related to both geography and diet. For example, the stable isotope ratios of *R. philippinarum* coincided with the topography of tidal flats, with evidence of significant differences among stations; a δ^15^N range was 11.7–12.9‰ along the inshore–offshore transect, despite the geographic vicinity within Japan [36]. However, regional variation is also possible; the Miya estuary of Japan had lighter values of δ^15^N (10.1 ± 0.4 ‰) [37]. The range of nitrogen values in France is 7.6–9.3‰ (mean 8.5 ± 0.5‰) [33], which is quite different from the findings of Watanabe et al. [37] and the present study.

These studies have demonstrated that a geographical profile and unique physiological status could be explained by the different metabolic pathways used by the clams and their populations, depending on environmental factors such as temperature, water mass, and food status, which could vary annually and seasonally. Regarding C and N, several studies have already shown the possibility that determining the isotope ratios of C and N is an easily applicable approach to identify the geographical origins of several types of seafood, including commercial hake and mackerel [15,32]. Komorita et al. also showed that the ratios of stable carbon and nitrogen isotopes in suspension-feeding bivalves typically reflect the values of consumed POM as they primarily feed on organic matter such as phytoplankton, detritus, and microphytobenthos suspended in water [2].

Although most studies have used isotopic carbon and nitrogen values to determine origins based on geography and diet, a dual plot of δ^13^C and δ^15^N values did not clearly differentiate the clams from the three countries in our study (Appendix A). The predictability of the LDA comprising the normalized values of these two elements (δ^13^C and δ^15^N) in the present study was also unsatisfactory (<70%) in terms of authenticating geographical origins (Table 2 and Table 3, and Figure 2A). Indeed, several items such as olive oil and wine, which have been studied extensively using C and N, might have more weighted information than aquatic organisms, since the carbon isotope ratios of olives or grapes can be quite clear, depending on their carbon metabolism (photosynthesis, Calvin cycle, etc.) and type of processing. Despite previous findings of pairs of C and N isotope ratios to determine geographic origins, the two factors might be insufficient to determine the origins of foods with limited information. However, C and N isotope ratios are useful when combined with other variables, other element profiles or metabolites determined using HR-MAS (high-resolution magic angle spinning) NMR [11,12,13]. Recent studies of isotopes for authentication have also suggested combining more than two elements to determine the origins of foods [14,17]. Thus, we suggest analyzing pairs of isotopes other than C and N to obtain supplementary information about the provenance of the Manila clam.

First, the stable isotope ratios of oxygen and hydrogen were added. Stable isotope ratios of oxygen and hydrogen in animal tissues and molecules are usually a function of environmental factors, such as water, and food influenced by geological factors such as temperature, precipitation, and surface water [39]. In general, oxygen and hydrogen stable isotope ratios that are dependent on latitude are well discussed, providing qualitative information about geographic origins [40], and can thus serve as natural tracers to follow migrations and authenticate regions [41].

Correlating temperature with ^18^O isotope values might be more informative [12,40], and previous studies of δ^18^O that have generally focused on inorganic tissues such as carbonates (shells) in clams have found that the oxygen isotope composition of mollusk shells (δ^18^O_shell_) provides reliable constraints on the temperature and oxygen isotope composition of the water (δ^18^O_w_) in which they were formed [42,43]. These findings showed that shell carbonate δ^18^O can serve as a temperature proxy, at least where δ^18^O remains constant and is known (δ^18^O_shell_ values might become depleted due to a higher pH, resulting in estimates of sea surface temperature that are up to 1 °C higher) [42]. Similarly, if the surface pH of the ocean falls due to a higher *p*CO_2_ for example, stable oxygen isotope fractionation between water and calcite would be greater, producing isotopically heavier shells, which could be wrongly interpreted as lower temperatures. Studies of food webs, however, have shown that the isotope values of hydrogen and oxygen can also be used as tracers of organismal food and resource use because the isotope compositions of these elements in tissues are influenced by diet as well as environmental waters [14]. These characteristics can also serve as markers in assessments of environmental differences among organisms to trace their geographical origins for food authentication [40]. However, although water is essential to aquatic organisms, stable isotopes of these elements have rarely been measured and applied to contemporary studies of animal trophic ecology. The present study showed that stable isotope ratios of oxygen in Manila clams slightly differed according to their authentications (*p* < 0.05, Table 1). The δ^18^O isotopic ratios of Manila clam samples from China, DPR Korea, and Korea were 22.0 ± 0.3‰, 23.0 ± 1.3‰ and 23.9 ± 2.2‰, respectively. The potential applications for determining nutrient sources with differences in hydrogen (δD) might also support distinct habitat information and thus, authentication [14]. The deuterium compositions (δD) of Manila clams from China and DPR Korea were −86.8 ± 4.9‰ and −81.3 ± 6.2‰, whereas that of the Korean samples was significantly lighter (−97.1 ± 10.6‰) (Table 1). Notably, δD and δ^18^O vary predictably in water according to water mass characteristics, decreasing from low-latitude, low-elevation coastal regions to high-latitude, inland, and mountainous regions [39]. In the context of animal tissues, δD and, to a lesser extent, δ^18^O are of increasing interest when linking spatial patterns of amount-weighted precipitation δD, and δ^18^O values (i.e., isoscapes) are a concern [14]. These values also facilitate tracking the movements of migratory animals and can be used in forensic approaches to animal- and plant-based materials. However, the present findings did not coincide with previous results of differences between oxygen and hydrogen isotope ratios associated with latitude. Rather, the interpretation of dietary differences is more plausible. According to Page et al. [4], the δD values of primary consumers can be used to assess basal resource contributions to the food web; for example, an increasing proportion of dietary algae in the gut reflects background isotope values more obviously. The contribution of anthropogenic and natural resources can directly connect with food webs. In addition, substantial and systematic differences in hydrogen isotope discrimination among non-macrophyte aquatic and terrestrial primary producers in freshwater and marine ecosystems can help discriminate between sources of organic matter and identify nutrient supplements [44]. Vander Zanden et al. [14] also showed that understanding controls on the isotopic composition of water in vivo (within consumers) is critical to accurately predict diet-tissue hydrogen and oxygen isotope offsets. For example, aquacultured oysters and clams with spatially variable diets and macroalgae as the dominant energy source can be distinguished [14]. This provides solid evidence that the composition of both hydrogen and the oxygen isotopes in clams is influenced not only by dietary, but also by body water isotope composition, and that these isotopes can also be used to trace and discriminate geographic origins based on organismal food and resource use. The composition of oxygen and deuterium in the study of clam samples was not always distinct, possibly because these offsets are water-dependent, and the sampling sites were geographically close Korea and DPR Korea.

When it comes to LDA using triple elements (C, N, and O or C, N, and D), the stable isotope ratio of oxygen or hydrogen could also better help to differentiate the environmental conditions of each site than the results from C-N pair trials, although the environmental parameters that can support the discriminant in isotope values could not be explained in our study (Table 3, Figure 2B,C). The findings of the quadruple LDA analysis comprising δ^13^C, δ^15^N, δD, and δ^18^O were more distinct than those of the triple isotope values (Table 3, Appendix A). This may seem like a conclusion that increasing the number of elements can yield better results in discrimination.

The results of adding only sulfur to a pair of C-N, however, showed more discriminatory results than those using C, N, O, and D in LDA, although it appeared that clams from DPR Korea and China were mixed (Table 3, Figure 2C). In this study, the Jinhae and Dolsan samples herein were characterized by a slightly higher δ^34^S compared with the Hwayang site in Korea. The average δ^34^S of the DPR Korean Manila clam was 21.89 ±0.90‰, which differed slightly from those of the samples from China (21.1 ± 0.5‰) and Korea (20.6 ± 0.2‰). In general, the geological effects on δ^34^S values in terrestrial plants, such as microbial processes in soil, fertilization procedures, and atmospheric deposition of sulfate aerosols in coastal areas mean that δ^34^S values can serve as trackers in studies of food web structures [45]. For example, stable S isotope ratios have been applied to estimate the relative proportions of foods derived from terrestrial and marine sources in the Inuit diet [46]. However, tracking relationships to the sources of foods can provide specific geographic source information. Thus, this method can be used to evaluate the geographic origins of aquaculture products [9]. In fact, with respect to aquaculture, fish migration has been investigated using δ^34^S values [47]. Due to the wide range of δ^34^S values in oceans and terrestrial landscapes and the influences of anthropogenic sources (for example coal burning), δ^34^S can distinguish the geographical origins of organisms by providing information about their habitats. Camin et al. [9] also noted that organic bound sulfur (S) in animal tissues can trace habitats because it is derived from organic S in dietary sources (such as plants). For this reason, the fact that clams are benthic organisms is also one of the reasons why δ^34^S values supported better discrimination than δ^18^O and δD, in our study. For the Manila clam in this study, combinations of three elements increased the discriminative power of the C-N combination in the order of O, D, and S (Table 3, Figure 2B–D). LDA analysis using four elements (C, N, S, and O or D) also showed higher discriminant power than the case using three elements excluding S (Figure 2E, Appendix A). Given that the *R. philippinarum* clam is benthic, geological effects such as redox conditions, microbial processes, and organic matter sulfurization recorded as δ^34^S values in clam tissues are decisive factors that can be plotted in adjacent areas in statistical analysis. Of course, the five-element LDA analysis (δ^13^C, δ^15^N, δ^18^O, δD, and δ^34^S) applied herein to Manila clams provided an even more vivid cluster, even in cross-validation analysis, which could distinguish samples based on their country of origin because they occurred in distinct clusters (except for two samples from China and DPR Korea that were very closely located) (Appendix A and Table 3). In brief, profiles could be distinguished more readily in clams by the combination of five (δ^13^C, δ^15^N, δ^18^O, δD, δ^34^S) rather than three elements in LDA.

It has been mentioned that other non-biological factors such as Sr, Nd, Pb, etc., may also be considered for food certification, just as sulfur showed information about the geological environment of clams [48]. Of them, strontium is not essential for life, but different Sr isotopes are associated with variations in the local geology of organisms [26]. Thus, ^87^Sr/^86^Sr ratios are vital geochemical tracers in several disciplines, including ecology, food sciences, archeology, and forensics. Applications of this element in ecology are based on the principle that Sr isotope ratios in organisms are not significantly affected by kinetic fractionations through biological processes and reflect sources of strontium that were available during the formation of these organisms [49]. The Sr isotope values of Manila clams from the three countries described herein were almost identical (*p* > 0.05) and close to that associated with older rocks (~0.710), whereas lower ratios (such as 0.702–0.706) indicated younger rocks such as volcanic basalt [26]. For example, ^87^Sr/^86^Sr ratios in rice from Japan, USA, China, and Thailand reflect the geochronological and lithological characteristics of the areas where they were grown [48]. Although the biological samples in the present study were formed in seawater, these values seemed to discriminate clam provenance for the same reason as δ^34^S, namely, geological information. Kennedy et al. also showed that Sr should not be ruled out in a combined isotope approach to fish migration studies in freshwater environments [50], because species in aquatic habitats close to land such as estuaries and coastal areas are directly affected by variations in Sr signatures.

Most chemical elements undergo isotopic fractionation in the natural environment during chemical transformations and changes in chemical status. Exceptions are radiogenic isotope systems such as ^143^Nd/^144^Nd (typically expressed using the epsilon notation εNd), which are conventionally normalized to a fixed stable isotope ratio and hence do not vary according to mass isotope fractionation. Consequently, these isotopic systems are particularly helpful for provenance studies, particularly those that trace the origin of water masses where living organisms might have formed. In contrast to Sr isotopes, the values of which are uniform in oceans, the radiogenic isotopic ratio of Nd varies widely in marine environments, mostly by reflecting the geological age of surrounding source rocks. We found that the individual Nd isotope values in the studied countries were highly variable and quite distinct (*p* < 0.05).

However, regarding the application of additional radiogenic isotope combinations, we confirmed that increasing the number of isotopes (elements) does not always improve the results of authentication studies (Appendix A). In the LDA analyses, Sr has more discriminative power than Nd (Figure 2E, Appendix A). Adding εNd even decreased the discrimination rate in the results of analyses using the elements, C, N, O, D, and S. Finally, LDA analyses that included a radiogenic isotope with C, N, S, O and C, N, S, O, D showed effective results. These findings indicated that more than five elements in the appropriate combination are required to evaluate the origins of the Manila clam (Figure 2F and Appendix A). Kasai et al. [1] found only small temporal changes in values of stable isotopes (except for a distinct decrease in those of carbon during heavy rain) in Manila clams, which proved the applicability of this technique to the validation of food sources. The present findings also emphasized the value of various stable isotopes (including δ^13^C and δ^15^N) to analyzing the provenance of food products, including Manila clams and other types of seafood. We found that the origins of Manila clams could be discriminated depending on geographic, climatic, and lithological differences. The flow diagram in Figure 2 shows that accumulated information about the geographic, climatic, diet, and lithological differences can help authenticate seafood origins. Moreover, our findings emphasized the potential value of combining Sr with the δ^13^C, δ^15^N, δ^18^O, δD, and δ^34^S isotopic signatures to progressively identify the origins of Manila clam samples. Therefore, such supportive information derived from more distinguished clustering of samples compared with other LDA analyses would help curtail seafood fraud and mislabeling. These results also suggested that the multi-element approach exerts a synergetic effect on authentication by enhancing explanatory power based on environmental factors. Furthermore, the cross-validated prediction accuracy, which provided greater confidence in the estimation test error, was lower than those from LDA in all cases (Table 3), but this was due to the small number of samples, and nonetheless the discriminant accuracy was more than 70% when applying the four elements. Analyzing multiple combined elements is a useful way to discriminate the origins of black tiger prawns and sea cucumbers based on the geographical properties of their specific habitats [11,12].

Finally, the fact that not all elements can increase the ability to discriminate the geographical origins of Manila clams suggests that further understanding of individuals must be considered when selecting appropriate elements for food authentication studies.

## 4. Conclusions

The quality of food products, especially their nutritional value, safety, and provenance, determines their consumer acceptance. Customer awareness has prompted regulatory authorities to prioritize appropriate labeling. Endeavors to curtail counterfeiting include testing products of suspect origin using sophisticated techniques. Here, we measured the stable isotope composition of main elemental constituents (C, N, H, O, and S) and radiogenic isotopic ratios (Sr, Nd) of bio-organic materials to improve the authentication and traceability of Manila clams collected in three adjacent countries. Our results emphasized the value of a multi-isotopic approach, as we showed that robust LDA analyses of five (δ^13^C, δ^15^N, δ^34^S, and δ^18^O plus δD) and six (δ^13^C, δ^15^N, δ^34^S, δ^18^O and δD, plus Sr or Nd) elements can determine the geographic origins of Manila clams. Although the values of cross-validation showed relatively low discriminant due to the fact that a small number of samples were used, recombination application and increasing resolution showed that the multi-isotope approach could provide reliable results for forensic applications and for authenticating the provenance of Manila clams. This present study validated the application of stable multi-element isotope analyses to food authentication and traceability and can help curb seafood fraud and mislabeling. Overall, the approach using isotopes of multiple elements unambiguously authenticated the provenance of commercially available Manila clams by revealing adulterations. Regional assignment of clam samples is most successful when multi-isotope values are combined in a multi-element LDA approach.

The Manila clam is one of the most consumed marine products in the world that is imported and produced locally. This means that this multi-proxy approach might have potential to be applied worldwide, and further study will be necessary. Furthermore, we also expect this method to be useful for other shellfish that are distributed in various regions and are imported and exported for figuring out their geographical origins.

## Figures and Tables

**Figure 1 foods-10-00646-f001:**
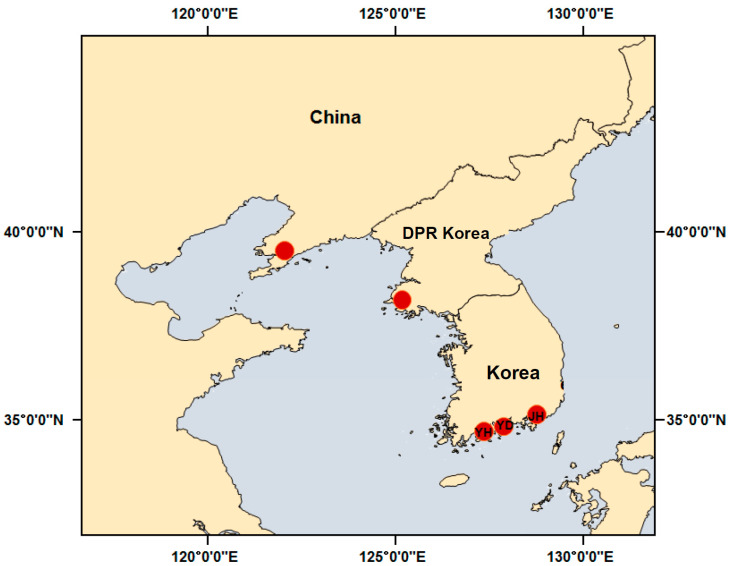
Geographic locations of sampling sites. China, Democratic People’s Republic of Korea (DPR Korea) and Korea.

**Figure 2 foods-10-00646-f002:**
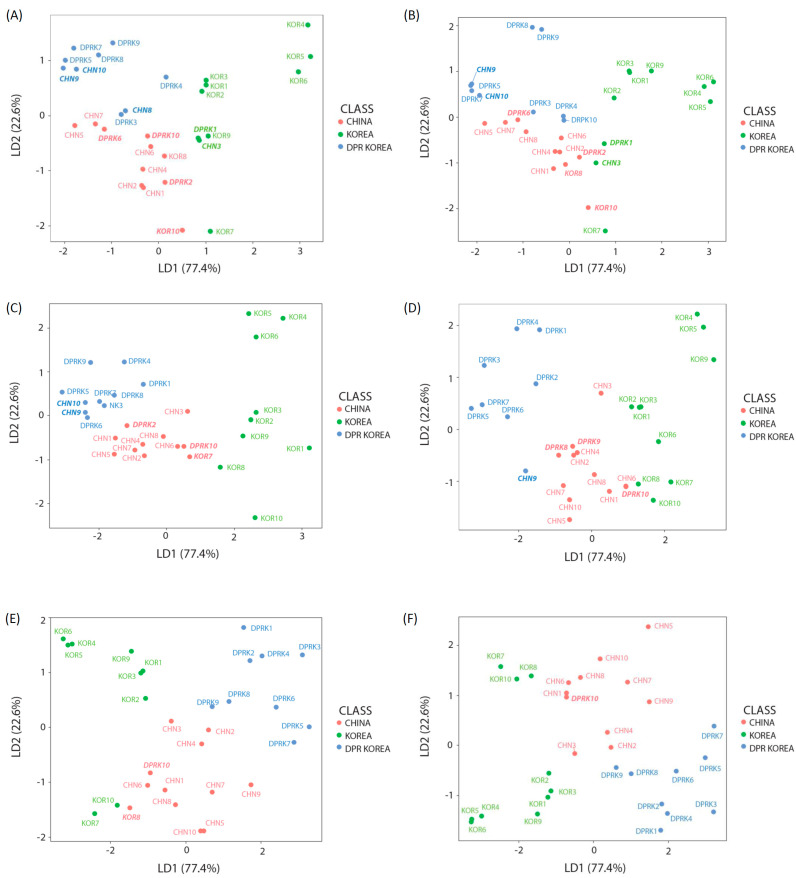
Manila clams from three countries grouped according to LDA findings of combined basic stable isotopes. (**A**) δ^13^C and δ^15^N-; (**B**)-δ ^13^C, δ^15^N, and δ^18^O-; (**C**) δ ^13^C, δ^15^N, and δD-; (**D**) δ^13^C, δ^15^N, and δ^3^S-; (**E**) δ ^13^C, δ^15^N, δ^34^S, and δ^18^O; and (**F**) δ ^13^C, δ^15^N, δ^34^S, δ^18^O, and Sr/Sr. Bold italic, which is shown as a different color for same country, indicates samples that have not been classified as the original provenance.

**Table 1 foods-10-00646-t001:** Isotope values of the Manila clam *Ruditapes philippinarum* from China (*n* = 10), DPR Korea (*n* = 10), and Korea (*n* = 10; three sites). Significant differences were analyzed by ANOVA (*p* < 0.05; Tukey post hoc tests). Superscript letters indicate significant difference following ANOVA (capitals and small letters represent significant differences in three countries and three sites in Korea, respectively).

Country	δ^13^C	δ^15^N	δ^18^O	δD	δ^34^S	^143^Nd/^144^Nd	^87^Sr/^86^Sr
China, Dalian (n = 10)	−17.7 (0.3) ^B,C^	8.8 (0.20) ^B,C^	22.0 (0.3) ^B,C^	−86.8 (4.9) ^B,C^	21.1 (0.5) ^A,C^	0.5119 (0.00006) ^A,B,C^	0.7094 (0.00011)
DPR Korea (n = 10)	−17.7 (0.4) ^B,C^	9.0 (0.2) ^A,B,C^	23.0 (1.29) ^A,B,C^	−81.3 (6.2) ^B,C^	21.9 (0.9) ^B^	0.5118 (0.0002) ^B,C^	0.7093 (0.000056)
Korea (n = 10)	−16.8 (0.4) ^A^	9.4 (0.6) ^A,B^	23.9 (2.2) ^A,B^	−97.1 (10.6) ^A^	20.6 (0.2) ^A,C^	0.5120 (0.0001) ^A,C^	0.7093 (0.000025)
-Yeosu Hwayang (YH) (n = 3)	−17.1 (0.02) ^a,b^	9.4 (0.05) ^a^	24.9 (0.7)	−107.1 (7.9) ^a,b^	20.7 (0.1) ^a,b,c^	0.5119 (0.00003) ^a^	0.7093 (0.000026)
- Yeosu Dolsan (YD) (n = 4)	−17.1 (0.2) ^a,b^	8.7 (0.3) ^b^	22.3 (2.8)	−96.8 (9.3) ^a,b,c^	20.7 (0.2) ^a,b^	0.5120 (0.00003) ^b^	0.7093 (0.000015)
- Jinhae (JH) (n = 3)	−16.3 (0.05) ^c^	10.1 (0.2) ^c^	25.2 (0.6)	−87.5 (4.8) ^b,c^	20.4 (0.1) ^a,c^	0.5121 (0.00003) ^c^	0.7093 (0.000023)

**Table 2 foods-10-00646-t002:** The data (stable isotope) set transformed and normalized by Z-score for statistical processing (linear discriminant analysis, LDA) in this study.

Sample	ID	δ^13^C	δ^15^N	δ^34^S	δ^18^O	δD	εNd	^87^Sr/^86^Sr
Korea	KR1	0.6539	0.8719	−0.6792	1.417	−2.805	−0.4344	−0.4868
	KR2	0.5998	0.7397	−0.5018	0.6678	−1.395	−0.09674	−0.6146
	KR3	0.6539	0.9381	−0.7385	1.411	−1.458	−0.4884	0.02455
	KR4	2.078	2.657	−1.009	1.137	−0.4215	1.227	−0.6146
	KR5	2.150	2.239	−0.9072	1.082	0.5346	1.598	−1.126
	KR6	1.988	1.908	−1.237	1.758	0.1487	1.261	−0.6146
	KR7	0.8883	−1.156	−0.3896	−1.652	0.5005	0.5738	−0.1307
	KR8	0.1310	−0.5168	−0.7679	−0.9099	−1.191	0.7136	−0.3590
	KR9	0.7440	0.1665	−0.7679	2.112	−1.024	1.018	−0.3590
	KR10	0.4736	−1.399	−0.3878	−1.184	−1.657	0.9838	−0.6146
China	CHN1	−0.1394	−1.156	−0.1628	−0.8429	1.121	−0.02921	−0.3590
	CHN2	−0.1665	−1.141	0.6598	−0.3590	0.3592	0.04729	−0.4625
	CHN3	0.6222	0.02967	0.5214	−0.7548	0.2213	−0.2804	−0.6719
	CHN4	−0.1711	−0.9010	0.4765	−0.5418	0.3978	0.004361	−0.1179
	CHN5	−1.213	−0.9114	−0.5979	−0.7958	−0.1410	−0.2603	4.073
	CHN6	−0.08372	−0.5090	−0.7320	−0.3374	−0.3770	−0.1837	1.471
	CHN7	−0.9140	−0.6960	−0.1665	−0.6114	−0.3036	−1.018	1.140
	CHN8	−0.4917	−0.2347	−0.5720	−0.8027	−0.5137	0.3004	0.09759
	CHN9	−1.456	−0.2082	−0.09640	−0.3861	0.2868	0.2342	−0.3590
	CHN10	−1.257	−0.09797	−0.9579	−0.6175	0.5296	0.2477	−0.1033
DPR Korea	NK1	0.6028	0.04868	1.962	−0.1968	1.310	−0.6541	0.2359
	NK2	0.1674	−0.8804	1.876	−0.3604	1.208	−0.5670	0.01865
	NK3	−0.5489	−0.3236	2.033	−0.1838	0.9059	−3.999	0.06630
	NK4	0.06214	0.6149	1.647	−0.3802	1.146	0.3288	−0.6583
	NK5	−1.430	−0.08148	1.194	−0.5140	0.9097	−0.3504	−0.8709
	NK6	−0.7793	−0.6909	1.336	−0.3869	1.050	−0.00767	−0.6713
	NK7	−1.323	0.1682	0.9653	−0.7450	0.08906	0.09464	1.353
	NK8	−0.9508	0.2988	−0.4765	1.502	0.1739	0.5651	−0.2311
	NK9	−0.7524	0.6074	−0.6792	1.417	0.9981	0.2949	−0.1033
	NK10	−0.1394	−0.3845	−0.8439	0.05863	−0.6029	−1.123	1.047

**Table 3 foods-10-00646-t003:** Numbers of individuals with geographical origins defined by LDA analyses of multi-element combinations. Each number indicates the samples whose origin was well identified out of 10 samples from each country. Numbers in parentheses represent cross-validation results of each geographical origin using leave-one-out cross-validation (LOOCV) method.

Combination	China (*n* = 10)	DPR Korea (*n* = 10)	Korea (*n* = 10)
C-N	6 (6)	6 (5)	8 (8)
C-N-O	7 (7)	7 (3)	8 (7)
C-N-D	8 (6)	8 (7)	9 (9)
C-N-S	9 (8)	7 (7)	10 (8)
C-N-O-D	8 (6)	8 (6)	9 (9)
C-N-D-S	9 (9)	7 (2)	9 (9)
C-N-S-O	10 (7)	9 (8)	9 (7)
C-N-S-Sr	9 (7)	7 (7)	10 (10)
C-N-S-O-D	9 (7)	9 (9)	10 (9)
C-N-S-O-Sr	10 (6)	9 (7)	10 (9)
C-N-S-D-Sr	9 (7)	7 (7)	9 (9)
C-N-O-D-Sr	6 (5)	8 (6)	9 (9)
C-N-O-D-Nd	8 (7)	8 (7)	9 (9)
C-N-S-O-D-Sr	10 (7)	9 (8)	10 (9)
C-N-S-O-D-Nd	9 (8)	9 (9)	9 (9)
C-N-S-O-D-Sr-Nd	10 (6)	9 (7)	10 (9)

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
