# Peer review of "A Multi-Elements Isotope Approach to Assess the Geographic Provenance of Manila Clams (Ruditapes philippinarum) via Recombining Appropriate Elements"

_foods, 2021, doi:10.3390/foods10030646_

Round 1
Reviewer 1 Report
This is the review of the manuscript titled “A multi-elements isotope approach to assess the geographic 2 provenance of Manila clams (Ruditapes phillippinarum): Extrac-3 tion of appropriate elements using recombination analysis “. The paper is very interesting and deal with a difficult subject. Nevertheless I have some overall concerns as per the presentation, the analysis and the overall way that this paper is written.
As I perceive, the aim of the paper is to develop a methodology/examine the possibilities, of on a multi-isotope approached in order to differentiate clams for three countries. In my opinion Figure S2 should be in the manuscript along with a new d18O vs dD diagrapm. Table S1 should also be in manuscript (maybe combined with Table 1). Contrary Table 2 should be in the sup. Inf. Figure 2 could be extended to include maybe all the combinations of elements as showed in sup. Figures S2-10. The discussion of pages 8 to 12 is very informative and demonstrates the difficulty in fingerprinting aquatic samples. Nevertheless some parts of it could be considered as introduction elements. In that sense figure S2 and a d18O vs dD diagram should be included and the analysis should be focused from these graphs and onward. For example, there is a clustering in China and N. Korea samples in these two diagrams that can be related maybe to proximity. Are any sea currents in the areas of Korean samples that may impose different temperatures? Furthermore, are there isoscapes of D and O of the area that have calculated a gradient of these isotopes ratios?
In my opinion, a reconstruction of the paper is needed as well as a major rewrite of pages 8-12. The subject and the multi-isotopes approach are extremely important. Please see in the attached pdf some additional minor comments

Author Response
We greatly appreciate critical comments of reviewers on our work and thank them for the valuable suggestions on the manuscript. According to reviewer’s comments, we thoroughly revised our manuscript and incorporated changes as suggested by the reviewer (please, check red color in the revised manuscript). We also answered one by one to comments raised by reviewers. Please attached file.
Reviewers' comments:

Reviewer 2 Report
This manuscript investigates the potential of using different combinations of 5 stable isotopes and two radiogenic elements to determine the geographic origin of Manila clams. This is a proof-of-concept study towards eventually using these tracers to determine the geographic origin of clams of unknown origin for food authentication purposes. I have a few questions about the methods and data interpretation (below), but overall the study seems well done. My biggest overall comment is that the authors only examined clams produced in North Korea, South Korea, and China, but the distribution of of R. philippinarum includes many other areas, such as Japan and the western coast of North America. Therefore, additional studies will be needed before this approach can be used more widely. There are some places where the English is unclear (some, but not all, are noted below).
Specific comments
Lines 20 and 24: Unclear what is meant by “recombination”.
Line 116-117: It isn’t clear to me whether the clams in this study were collected from the wild or whether they were farmed raised. Please clarify. Also, please specify the time of year these samples were collected.
Lines 120-121: I don’t understand what is meant by “The sites of China and North Korea refer to regions directly imported from NIFS.”
Line 141: What “standards” were used?
Lines 131-150: More details about how the stable isotope data were normalized and what standards were used (including their accepted isotope values) is needed.
Page 8, first paragraph: Please cite Table 1 and Fig. S1 when presenting these results.
Page 8, first paragraph: Table 1 says that the d13C value for North Korea is -17.7 +/- 0.37 per mil. Which is correct?
Page 8, second paragraph: Table 1 says that the d15N value for South Korea is 9.35 +/- 0.63 per mil. Which is correct?
LDA figures: I don’t understand why some data points that are the color of one country have a text label for a different country. Are these samples that were grouped incorrectly? Perhaps it would be helpful if the authors briefly described how to interpret the LDA plots.
Table 2: Please clarify what the numbers in this table indicate.
Page 11, second full paragraph: Table 2 says “C-N-O-H” whereas this text says “C-N-O-D”. Please be consistent.
Figs S7 and S10: I don’t believe these figures are referenced in the text.
Author Response
We greatly appreciate critical comments of reviewers on our work and thank them for the valuable suggestions on the manuscript. According to reviewer’s comments, we thoroughly revised our manuscript and incorporated changes as suggested by the reviewer (please, check red color in the revised manuscript). We also answered one by one to comments raised by reviewers. Please attached file.

Round 2
Reviewer 1 Report
Dear Authors,
My concerns on the manuscript were fully addressed. I recommend publishing as is.
Best regards.